# Root microbiota analysis of *Oryza rufipogon* and *Oryza sativa* reveals an orientation selection during the domestication process

Liqun Jiang,[1] Da Ke,[1] Bingrui Sun,[1] Jing Zhang,[1] Shuwei Lyu,[1] Hang Yu,[1] Pingli Chen,[1] Xingxue Mao,[1] Qing Liu,[1] Wenfeng Chen,[1] Zhilan Fan,[1] Li Huang,[2] Sanjun Yin,[2] Yizhen Deng,[3] Chen Li[1]

**ABSTRACT** The root-associated microbiota has a close relation to the life activities of plants, and its composition is affected by the rhizospheric environment and plant genotypes. Rice (*Oryza sativa*) was domesticated from the ancestor species *Oryza rufipogon*. Many important agricultural traits and adversity resistance of rice have changed during a long time of natural domestication and artificial selection. However, the influence of rice genotypes on root microbiota in important agricultural traits remains to be explained. In this study, we performed 16S rRNA and internal transcribed spacer (ITS) gene amplicon sequencing to generate bacterial and fungal community profiles of *O. rufipogon* and *O. sativa*, both of which were planted in a farm in Guangzhou and had reached the reproductive stage. We compared their root microbiota in detail by alpha diversity, beta diversity, different species, core microbiota, and correlation analyses. We found that the relative abundance of bacteria was significantly higher in the cultivated rice than in the common wild rice, while the relative abundance of fungi was the opposite. Significant differences in agricultural traits between *O. rufipogon* and *O. sativa* showed a high correlation with core microorganisms in the two *Oryza* species, which only existed in either or had obviously different abundance in both two species, indicating that rice genotype/phenotype had a strong influence on recruiting specific microorganisms. Our study provides a theoretical basis for the in-depth understanding of rice root microbiota and the improvement of rice breeding from the perspective of the interaction between root microorganisms and plants.

**IMPORTANCE** Plant root microorganisms play a vital role not only in plant growth and development but also in responding the biotic and abiotic stresses. *Oryza sativa* is domesticated from *Oryza rufipogon* which has many excellent agricultural traits especially containing resistance to biotic and abiotic stresses. To improve the yield and resistance of cultivated rice, it is particularly important to deeply research on differences between *O. sativa* and *O. rufipogon* and find beneficial microorganisms to remodel the root microbiome of *O. sativa*.

**KEYWORDS** root-associated microbiota, *Oryza rufipogon*, *Oryza sativa*, ITS and 16S rRNA, diversity analysis

Address correspondence to Chen Li, lichen@gdaas.cn, or Yizhen Deng, dengyz@scau.edu.cn.

Liqun Jiang and Da Ke contributed equally to this article. Author order was determined by decreasing contribution.

The authors declare no conflict of interest.

See the funding table on p. 13.

Plant root microorganisms, including bacteria, fungi, actinomycetes, and protozoa (1, 2) play a vital role not only in plant growth and development (3, 4–8) but also in responding the biotic and abiotic stresses (3, 6, 9, 10). For example, in *Arabidopsis thaliana*, root microorganisms regulate both phosphorus stress tolerance and immunity response based on jasmonic acid and salicylic acid pathway toward *Hyaloperonospora arabidopsidis* and *Pseudomonas syringae* (11). In return, plant roots secrete metabolites,

which are not only essential substances for root bacterial survival and activity but also recruit beneficial microorganisms and inhibit some plant pathogens (12, 13). However, few successful cases occurred in directly utilizing the activity of root microbiota to improve the plant production systems except for the symbiotic relationship between arbuscular mycorrhizal fungi and leguminous plants (14). Therefore, it is of great value to reveal the relationship between root microbiota structure and function, in order to promote highly efficient utilization in green agriculture (15).

Rice is one of the most important cereal crops all over the globe, which feeds more than half the population of the world (16) and China contributes nearly 30% of global rice production (17). During a long time of natural evolution and artificial selection, the cultivated rice (*Oryza sativa*) evolves from wild rice (*Oryza rufipogon*), with a significant change in phenotypes of the important agronomic traits and resistance to the biotic and abiotic stresses (18, 19). From the view of rice domestication, root microbiota resources that co-evolved and interacted with the host over a long time may benefit to efficiently improve crop production (20, 21). For example, *Rhizobium* spp., *Pseudomonas* spp., and *Bacillus subtilis* FB17 produced signals to induce plant defense against pathogens and promoted plant growth and development by regulating the metabolism of carbon and nitrogen (21, 22).

Although some root microorganisms have been reported to be beneficial for promoting crop production, different kinds of microorganisms regulating other important agronomic traits also need to be found and studied. Deep research on the microbiome of various plants benefited from fast-growing high-throughput amplicon sequencing techniques, such as *A. thaliana* (23–26), *O. sativa* (27–29), *Triticum aestivum* (30–32), *Zea mays* (33), *Solanum lycopersicum* (34, 35), *Setaria italica* (36), and *Malus pumila* (37). Previous research indicated that plant root microbiota mainly consisted of fungi *Ascomycota*, *Basidiomycota*, *Crytridiomycota*, and *Rozellomycota* (38), and bacteria *Actinobacteria*, *Bacteroidetes*, *Firmicutes*, and *Proteobacteria* (39, 40). The composition of root microbiota is influenced not only by soil conditions, such as pH value, water content, and relative abundance of carbon and nitrogen (41–43) but also by different plant species (12) or different genotypes in the same species (3).

Compared with cultivated rice, the common wild rice shows creep growth, divergent fringe, longer awn, stronger dormancy, easier shattering, and has a stronger ability to survive in hostile environments may be because of some special root microorganisms that provide essential function of fighting against pathogens, absorbing nutrients, and responding to abiotic stress in order to improve the resistivity to adversity and promote host healthy development (44–46). However, some root microorganisms were lost during the domestication process, so we need to explore the common and the different microbiota between *O. sativa* and *O. rufipogon* in order to find and utilize the beneficial microorganisms to improve rice production and quality. In this study, we used the amplicon sequencing technique [16S rRNA and internal transcribed spacer (ITS)] to analyze the population structure, diversity, and assembly process of symbiotic microorganisms between *O. rufipogon* and *O. sativa*. We independently and contrastively analyzed the correlation of bacterial and fungal microbiota in each and both of the two *Oryza* species and established the core microorganisms. We also evaluated the correlation between the different agronomic traits and different abundance of microbiota in the two *Oryza* species. This study could not only provide different or new microorganisms as supplementary resources to improve rice production or resistance to adversity but also put forward a new insight into rice domestication based on the change in microbiota from *O. rufipogon* to *O. sativa*.

## MATERIALS AND METHODS

### Experimental materials and sampling

In this study, 10 common wild rice accessions (Or01 to Or10) and 10 cultivated rice varieties (Os01 to Os10) were, respectively, collected in the paddy field in Lyutian town, Conghua district, Guangzhou city, Guangdong province, China. Root samples with adherent soil were collected from each rice plant (Table S1). For consistency, 10 cm of root was cut off by scissors from each rice plant and immediately washed in 40 mL PBS buffer (pH 7.0, per liter 16.5 g of $Na_2HPO_4 \cdot 7H_2O$, 6.33 g of $NaH_2PO_4 \cdot H_2O$, 200 mL Silwet L-77) shaking three times at 180 r/min, followed by blotting up water attached on the root and frozen in liquid nitrogen for storing at $-80°C$ (1).

### DNA extraction and high-throughput sequencing

Total genome DNA from samples were extracted using the E.Z.N.A. Soil DNA Kit (Omega Bio-tek, Norcross, GA, USA) and, respectively, checked by 1% agarose gel electrophoresis for purity detection and ultraviolet spectrophotometry for concentration detection. DNA was diluted to 1 ng/uL using sterile water for amplification. 16S rRNA and ITS genes were, respectively, amplified using the specific primers with the barcode. The specific primers 515F (forward primer, 5′-GTGCCAGCMGCCGCGG-3′) and 806R (reverse primer, 5′-GGACTACHVGGGTWTCTAAT-3′) with barcode were used for bacterial 16S rRNA gene tags (V4 region) amplification, while the specific primers 1723F (forward primer, 5′-CTTG GTCATTTAGAGGAAGTAA-3′) and 2043R (reverse primer, 5′-GCTGCGTTCTTCATCGATGC-3′) were used for fungal ITS gene (ITS1 region) amplification. PCR amplification was carried out with Phusion High-Fidelity PCR Master Mix (New England Biolabs) consisting of initial denaturation at 98°C for 1 min, followed by 30 cycles of denaturation at 98°C for 10 s, annealing at 50°C for 30 s, and elongation at 72°C for 30 s, and then a final extension at 72°C for 5 min. PCR products were checked by 2% agarose gel electrophoresis for mixing the same volume of 1× loading buffer (contained SYB green). Samples with bright main strips between 400 and 450 bp were chosen to be mixed in equidensity ratios and purified using the AxyPrep DNA Gel Extraction Kit (Axygen Biosciences, Union City, CA, USA) according to the manufacturer's instructions and quantified using QuantiFluor-ST (Promega, USA). Sequencing libraries were produced by using the Illumina TruSeq DNA PCR-Free Library Preparation Kit (Illumina, San Diego, CA, USA) following the manufacturer's instructions, and index codes were added. The library quality was assessed on the Qubit@ 2.0 Fluorometer (Thermo Scientific) and Agilent Bioanalyzer 2100 system. The library was sequenced on an Illumina Novaseq platform and 250 bp paired-end reads were generated.

### Bioinformatics analysis

After removing the adaptors, primers, and low-quality reads, the pair-end reads were overlapped to assemble the final sequences. The criterion of overlapping was the overlapping region lengths larger than 10 bp and a mismatch ratio lower than 0.2. Chimera tags were filtered out using the Gold database by UCHIME (version 4.2.40). Then operational taxonomic unit (OTU) analysis was performed using the Uparse package (version 7.0.1001) with a 97% sequence identity. Each OTU was taxonomically assigned to the silva database using the Ribosomal Database Project (RDP) classifier OTUs were processed by removing chloroplast sequences, chondriosome sequences, and unclassified sequences. The OTUs with relative abundance values >0.001% (above three tags in at least one sample) in at least one sample were retained. In order to compare the samples at the same sequencing depth, we selected 35,000 sequences obtained by random sampling.

### Alpha and beta diversity analysis

In-house Perl scripts were used to analyze alpha (within samples) and beta (among samples) diversity. In order to compute alpha diversity, we analyzed the OTU table

and calculated three indexes: Chao1, Observed species, and PD whole tree. Chao1 can reflect the community abundance; Observed species indicated the number of OTU; and PD whole tree reflects the total represent sequences of OTU. Beta diversity analysis was conducted to examine the similarity of the community structure among different samples. The principal coordinates analysis (PCoA) was performed on the community composition structure at the genus level to explore the similarities or dissimilarities between *O. rufipogon* and *O. sativa*, which was applied to reduce the dimension of the original variables using the QIIME software package. Cluster analysis mainly refers to the hierarchical clustering analysis method using any distance to evaluate the similarity between the two *Oryza* species.

## Statistical test

A permutational multivariate analysis of variance was performed using QIIME software and 999 displacement tests to determine the differences between bacteria and fungi in the two *Oryza* species. Redundancy analysis was analyzed using R to figure out the relationships between soil microorganisms and agricultural traits.

## RESULTS

### High-throughput amplicon sequencing and relative abundance analysis

After paired-end alignments and quality filtering, a total of 674,923 bacterial 16S and 741,398 fungal ITS taxon tags were recovered and assigned to 80,178 bacterial and 11,078 fungal OTUs, respectively (Table S2). Relative abundance analysis of bacteria based on taxonomic phylum between *O. rufipogon* and *O. sativa* showed that all of the five phyla, *Proteobacteria*, *Thaumarchaeota*, *Bacteroidetes*, *Chloroflexi,* and *Acidobacteria*, were top five of relative higher abundance both in the two *Oryza* species, although relative abundance of *Proteobacteria* and *Bacteroidetes* were higher in *O. sativa* while *Thaumarchaeota* was higher in *O. rufipogon* (Fig. 1A). The same analysis for fungi showed that *Ascomycota*, *Basidiomycota*, *Chytridiomycota*, *Mortierellomycota,* and *Rozellomycota* were the top five phyla with relative higher abundance, in which total of relative abundance of *Ascomycota* and *Basidiomycota* accounted for more than 70% both in the two *Oryza* species, although relative abundance of *Basidiomycota* was slightly higher and *Ascomycota* was lower in *O. sativa* (Fig. 1B). In addition, contrastive analysis of relative abundance of top 35 fungi and bacteria based on taxonomic genus between *O. rufipogon* and *O. sativa* were completed. As shown in Fig. S1A, eight bacterial genera had relatively higher abundance both in the two *Oryza* species, including *Candidatus Solibacter*, *Pseudomonas*, *Geobacter*, *Acinetobacter*, *Candidatus Nephrothrix*, *Anaeromyxobacter*, *Bryobacter,* and *Stenotrophomonas*, while the relative abundance of *Pseudomonas* was higher in *O. sativa* than in *O. rufipogon*. The relative abundance of the top 35 genera accounted for only 10% indicating that bacterial species were very rich both in the two *Oryza* species. For fungi, eight genera had relatively higher abundance both in the two *Oryza* species, including *Conocybe*, *Westerdykella*, *Echria*, *Mortierella*, *Hebeloma*, *Fusarium*, *Clitopilus,* and *Panaeolus*. The relative abundance of *Mortierella* was significantly lower in *O. sativa*, which resulted as the dominant fungal genus in *O. rufipogon*, while *Conocybe*, *Echria,* and *Fusarium* had an obviously higher abundance in *O. sativa*, and the relative abundance of *Conocybe* was more than 20% to be the dominant fungal genus (Fig. S1B).

### Diversity analysis of root microbial community between *O. rufipogon* and *O. sativa*

We used the Chao1 index and Observed species index to analyze the richness and the PD whole tree index to analyze the diversity of root microbial communities between *O. rufipogon* and *O. sativa* at the OTUs level. As shown in Fig. 2A, for the bacterium, significant enhancements in the Chao1 index ($P = 0.0343$) and Observed species index ($P = 0.0343$) indicated the richness in *O. sativa* was significantly higher than in *O. rufipogon*, and PD whole tree index ($P = 0.0126$) indicated a same tendency for the diversity.

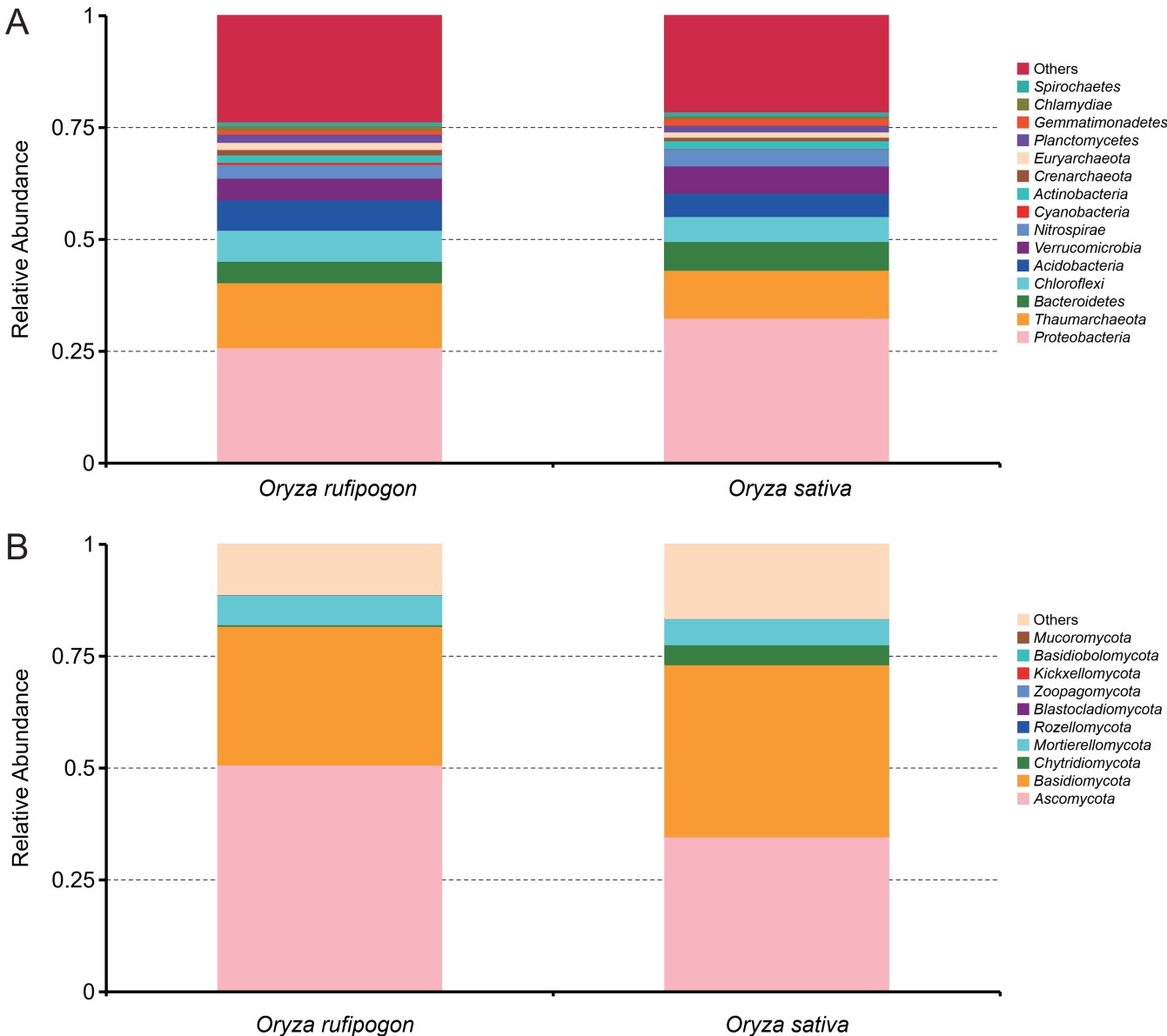

**FIG 1** Relation abundance of bacterial and fungal composition at phyla level of *O. rufipogon* and *O. sativa*. (A) Bacteria. (B) Fungi.

Interestingly, an opposite variation of PD whole tree index ($P = 0.0126$) for fungi showed that the diversity in *O. rufipogon* was significantly higher than in *O. sativa*, although both the Chao1 index ($P = 0.45$) and Observed species index ($P = 0.496$) showed only slightly higher richness in *O. rufipogon* (Fig. 2B).

In order to observe the similarities and dissimilarities of fungi and bacteria between *O. rufipogon* and *O. sativa*, PCoA was performed based on Bray–Curtis. As shown in Fig. 3a, PCoA showed distinct differences in microbial communities existed between *O. rufipogon* and *O. sativa*, with the two principal components (PC1 and PC2) of PCoA explained 26% and 14% (bacteria, Fig. 3A), 33% and 15% (fungi, Fig. 3B), respectively. Analysis of Similarities (ANOSIM) and Multiple Response Permutation Procedure (MRPP) analysis (Table 1) showed that it is more similar to each other within species than that between the two *Oryza* species, for both bacteria (ANOSIM: R = 0.514, $P = 0.001$ and MRPP: A = 0.08, $P = 0.001$) and fungi (ANOSIM: R = 0.799, $P = 0.001$ and MRPP: A = 0.16, $P = 0.001$).

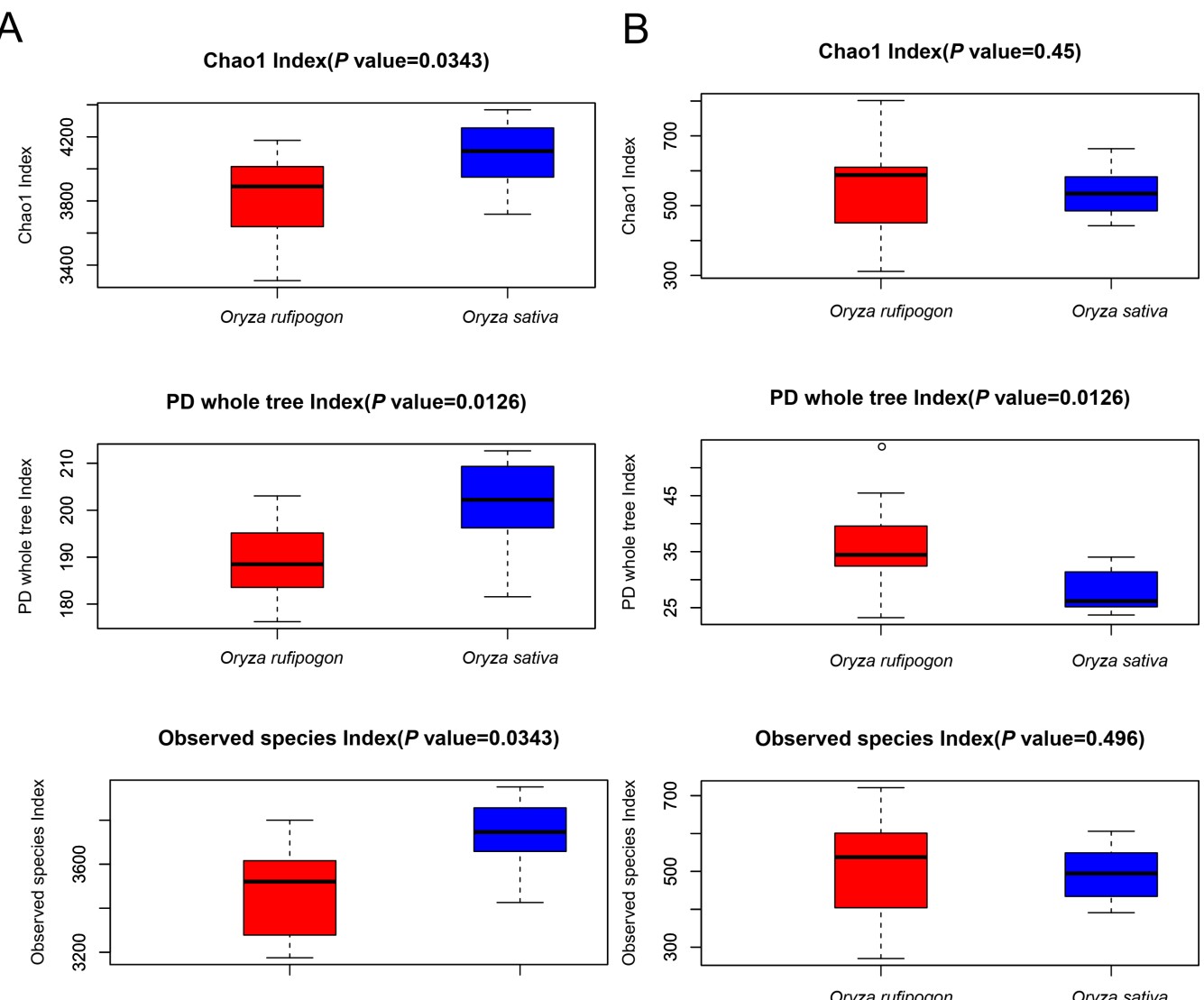

**FIG 2** Root bacterial and fungal diversity indices of *O. rufipogon* and *O. sativa* (Chao1, PD whole tree, and Observed species) calculated by 16S rRNA and ITS gene sequence data at OTU level. (A) Bacteria. (B) Fungi.

## Difference and similarity analysis of root microbial community between *O. rufipogon* and *O. sativa*

We further analyzed the different and similar microbial species between *O. rufipogon* and *O. sativa* based on OTUs. As shown in Venn graph (Fig. S2), 3,422 and 3,700 bacterial OTUs were, respectively, specially enriched around the roots of *O. rufipogon* and *O. sativa*, while 7,474 OTUs were commonly enriched in the two *Oryza* species (Fig. S2A). For fungi, 717 and 628 OTUs were, respectively, specially enriched around the roots of *O. rufipogon* and *O. sativa*, while 952 OTUs were commonly enriched in the two *Oryza* species (Fig. S2B).

**TABLE 1** ANOSIM and MRPP analysis of bacteria and fungi between *O. rufipogon* and *O. sativa*

| | ANOSIM analysis | | MRPP analysis | | | |
|---|---|---|---|---|---|---|
| *O. rufipogon* vs *O. sativa* | R-value | *P*-value | A | Observed-delta | Expected-delta | Significance |
| Bacteria | 0.5140 | 0.0010 | 0.0824 | 0.4284 | 0.4669 | 0.0010 |
| Fungi | 0.7987 | 0.0010 | 0.1624 | 0.6306 | 0.7528 | 0.0010 |

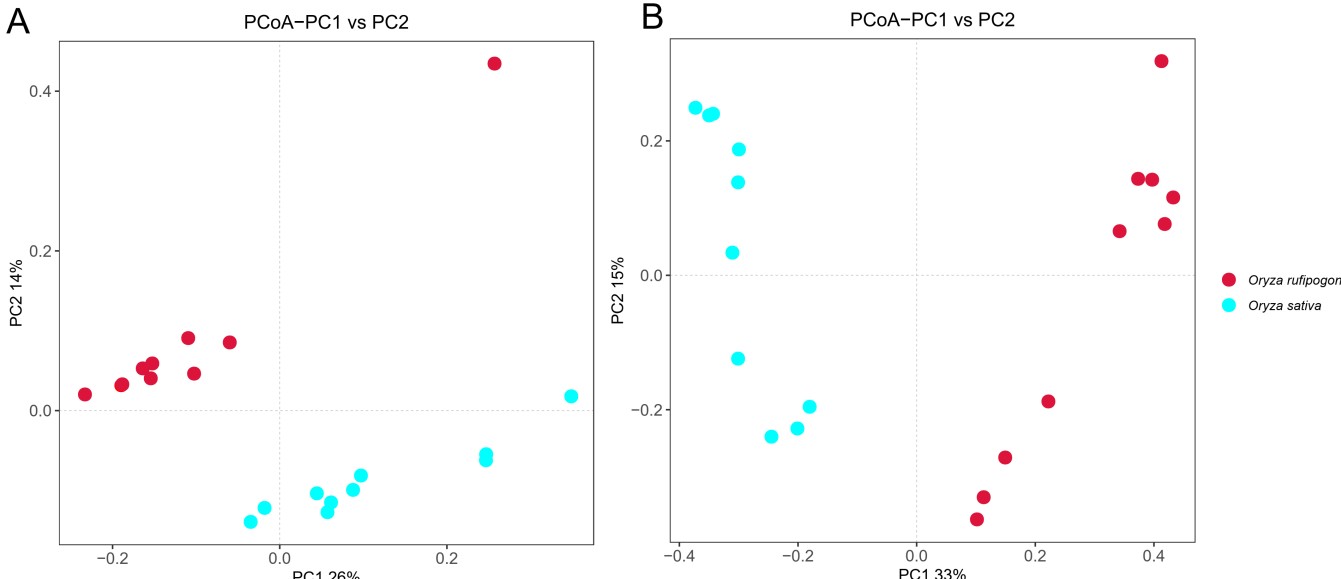

**FIG 3** PCoA based on Bray–Curtis dissimilarity metrics, showing the distance in the bacterial and fungal communities between *O. rufipogon* and *O. sativa*. (A) Bacteria. (B) Fungi.

Different bacterial OTUs between *O. rufipogon* and *O. sativa* were classified into 20 genera (Table 2) while differentially enriched fungal OTUs were classified into 13 genera (Table 3). Sixteen genera of bacteria (*Pseudomonas*, *Acinetobacter*, *Stenotrophomonas*, *Duganella*, *MDN1*, *Sphingobacterium*, *Nitrospira*, *Ellin6067*, *Dyella*, *Bdellovibrio*, *Terrimonas*, *Aquicella*, *Massilia*, *Pelobacter*, *Sphingomonas,* and *Thiobacillus*) showed relative higher abundance in *O. sativa* than in *O. rufipogon*, while four genera (*Bryobacte*, *Bradyrhizobium*, *Denitratisoma,* and *Opitutus*) were the opposite. For fungus, relative abundance of 10 genera (*Conocybe*, *Echria*, *Clonostachys*, *Sarocladium*, *Psathyrella*, *Parasarocladium*, *Trichoderma*, *Zopfiella*, Metapochonia, and *Limnoperdon*) were significantly higher in *O. sativa* than in *O. rufipogon*, while three genera (*Hebeloma*, *Clitopilus,* and *Neurospora*) showed the opposite trend.

## Correlation analysis of root-associated microbiome of *O. rufipogon* and *O. sativa*

We analyzed the correlation between bacteria and bacteria, fungi and fungi, and bacteria and fungi based on the common genera of the top 35 genera in *O. rufipogon* and *O. sativa*. Twenty-four of the top 35 bacterial genera commonly existed in both *O. rufipogon* and *O. sativa*, while 28 common fungal genera existed in the two *Oryza* species (Fig. 4). Observed positive correlation among microorganisms indicated a mutual synergistic effect, while significant negative correlation among microorganisms showed an interactive antagonistic effect.

As shown in Fig. 4A and B, comparative analysis of correlation among bacterial genera in *O. rufipogon* and *O. sativa* showed the obvious positive correlation between *Phenylobacterium* and *Rhodomicrobium*, and both *Geothrix* and *Geobacter* with either *Novosphingobium* or *Anaeromyxobacter* in the two *Oryza* species, while only significant negative correlation between *Occallatibacter* and *Desulfovirga*. Correlation between *Anaeromyxobacter* and either *Geobacter* or *Spirochaeta*, *Ellin6067,* and *Granulicella* were observed positive in *O. rufipogon* but significantly negative in *O. sativa*.

In both *O. rufipogon* and *O. sativa*, there was a noted positive correlation among certain fungal genera. Specifically, Sarocladium, Fusarium, and Microdochium exhibited positive correlations, as did Didymella, Nigrospora, Cladosporium, and Clonostachys, except for the correlation between Cladosporium and Nigrospora. Additionally, positive correlations were observed among Myxospora, Edenia, and Myrmecridium, as well as

**TABLE 2** Difference analysis based on bacterial genus between *O. rufipogon* and *O. sativa*

| Comparison | Taxon | Mean (*O. rufipogon*) | Mean (*O. sativa*) | *P*-value | Q-value | Difference |
|---|---|---|---|---|---|---|
| *O. rufipogon* vs *O. sativa* | *Pseudomonas* | 0.0003 | 0.0055 | 0.001 | 0.0111 | Up |
| | *Acinetobacter* | 0.0002 | 0.0034 | 0.05 | 0.123 | Up |
| | *Bryobacter* | 0.0085 | 0.0056 | 0.024 | 0.0797 | Down |
| | *Stenotrophomonas* | 0.0002 | 0.003 | 0.002 | 0.0183 | Up |
| | *Duganella* | 0.0006 | 0.0027 | 0.017 | 0.0645 | Up |
| | *MND1* | 0.0028 | 0.0065 | 0.001 | 0.0111 | Up |
| | *Bradyrhizobium* | 0.0044 | 0.0027 | 0.002 | 0.0183 | Down |
| | *Sphingobacterium* | 0.0001 | 0.0015 | 0.005 | 0.0309 | Up |
| | *Denitratisoma* | 0.003 | 0.0009 | 0.013 | 0.0576 | Down |
| | *Nitrospira* | 0.0013 | 0.0035 | 0.001 | 0.0111 | Up |
| | *Ellin6067* | 0.002 | 0.0037 | 0.004 | 0.0287 | Up |
| | *Dyella* | 0.0004 | 0.0014 | 0.023 | 0.0773 | Up |
| | *Bdellovibrio* | 0.0006 | 0.0021 | 0.003 | 0.0242 | Up |
| | *Terrimonas* | 0.0004 | 0.0032 | 0.001 | 0.0111 | Up |
| | *Aquicella* | 0.0003 | 0.001 | 0.002 | 0.0183 | Up |
| | *Massilia* | 0.0004 | 0.0015 | 0.005 | 0.0309 | Up |
| | *Opitutus* | 0.0017 | 0.001 | 0.004 | 0.0287 | Down |
| | *Pelobacter* | 0.0006 | 0.0013 | 0.011 | 0.0531 | Up |
| | *Thiobacillus* | 0.0004 | 0.0012 | 0.001 | 0.0111 | Up |
| | *Sphingomonas* | 0.0003 | 0.0011 | 0.001 | 0.0111 | Up |

between Pseudeurotium and either Talaromyces or Coniochaeta, in both *O. rufipogon* and *O. sativa*. On the other hand, positive correlations among Myrmecridium, Acremonium, Talaromyces, Boothiomyces, Panaeolus, and Pyrenochaetopsis were noted, except for the correlation between Boothiomyces and Talaromyces. Furthermore, positive correlations among Trichoderma, Neurospora, and Westerdykella were found exclusively in *O. rufipogon*. Compared with the microorganisms with observed positive correlation, the population of microorganisms with significant negative correlation was relatively less, for only *Conioscypha* and *Psilocybe* showing significant negative correlation in *O. rufipogon* and a few in *O. sativa*, such as *Echria* with either *Cladosporium*, *Clonostachys*, *Edenia,* or *Pyrenochaetopsis* (Fig. 4C and D).

A combination correlation analysis between bacteria and fungi showed that only observed positive correlation between *Bryobacter* and *Clitopilus* existed in both *O. rufipogon* and *O. sativa*, indicating that they are synergistic. Most correlations between bacteria and fungi in *O. rufipogon* showed no significant correlation in *O. sativa*, and vice versa. For example, in *O. rufipogon*, three bacterial genera (*Ellin 6067*, *Anaerolinea,* and

**TABLE 3** Difference analysis based on fungal genus between *O. rufipogon* and *O. sativa*

| Comparison | Taxon | Mean (*O. rufipogon*) | Mean (*O. sativa*) | *P*-value | Q-value | Difference |
|---|---|---|---|---|---|---|
| *O. rufipogon* vs *O. sativa* | *Conocybe* | 0.0186 | 0.2219 | 0.001 | 0.0093 | Up |
| | *Echria* | 0.0016 | 0.0494 | 0.002 | 0.0151 | Up |
| | *Hebeloma* | 0.0107 | 7.45E-05 | 0.001 | 0.0093 | Down |
| | *Clitopilus* | 0.0224 | 0.0035 | 0.002 | 0.0151 | Down |
| | *Clonostachys* | 0.0013 | 0.0077 | 0.01 | 0.0447 | Up |
| | *Sarocladium* | 0.0008 | 0.0044 | 0.009 | 0.0418 | Up |
| | *Neurospora* | 0.005 | 0.0005 | 0.001 | 0.0093 | Down |
| | *Psathyrella* | 2.86E-06 | 0.0014 | 0.006 | 0.0329 | Up |
| | *Parasarocladium* | 0.0002 | 0.0021 | 0.044 | 0.1398 | Up |
| | *Metapochonia* | 0.0003 | 0.0014 | 0.007 | 0.367 | Up |
| | *Trichoderma* | 0.0008 | 0.0029 | 0.001 | 0.0093 | Up |
| | *Limnoperdon* | 0.0001 | 0.0011 | 0.026 | 0.0923 | Up |
| | *Zopfiella* | 0.0002 | 0.0026 | 0.001 | 0.0093 | Up |

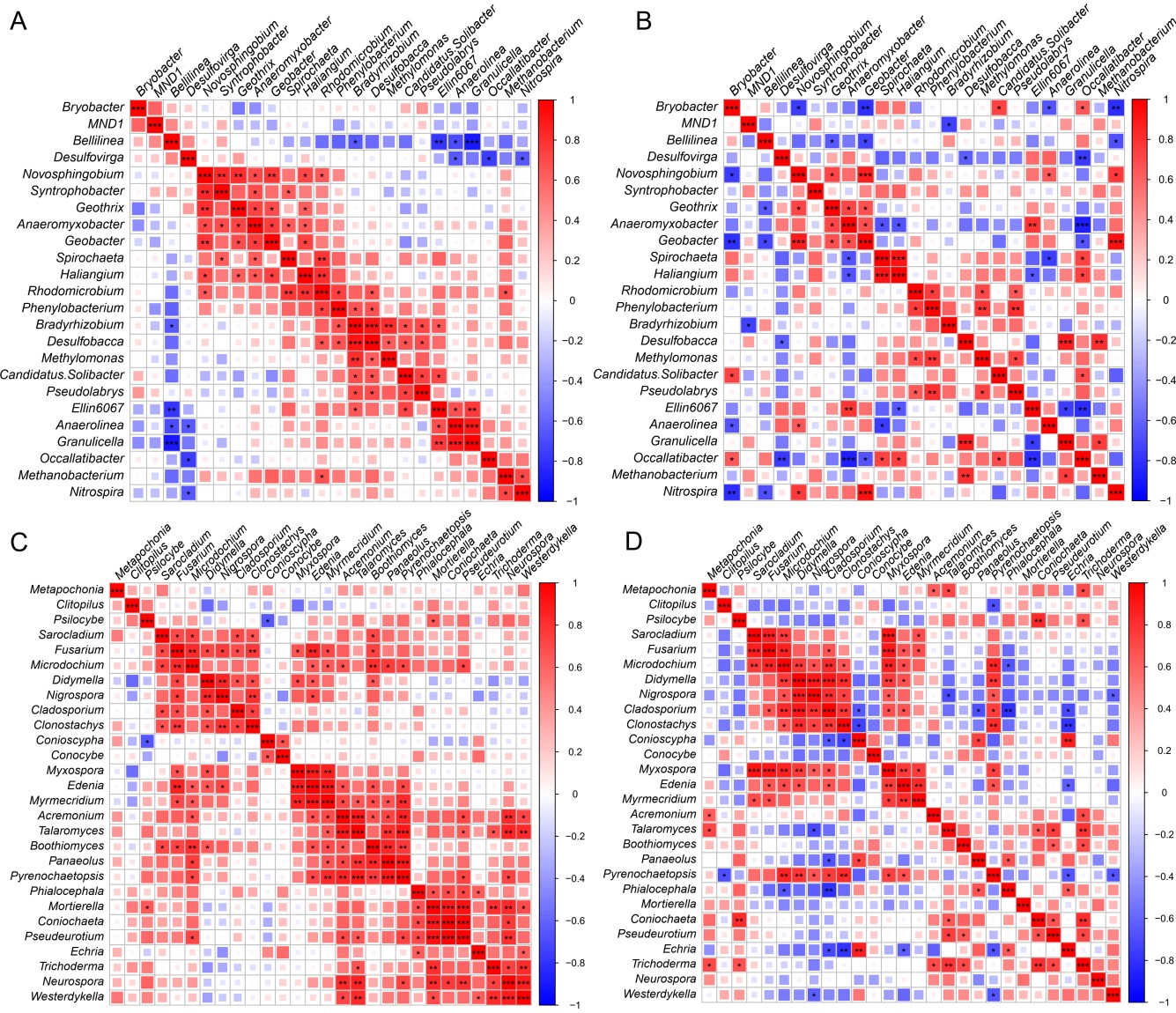

**FIG 4** Spearman correlation of bacteria–bacteria, fungi–fungi at genera level in *O. rufipogon* and *O. sativa*. (A) Correlation among bacteria in *O. rufipogon*. (B) Correlation among bacteria in *O. sativa*. (C) Correlation among fungi in *O. rufipogon*. (D) Correlation among fungi in *O. sativa*. Red represents a positive correlation, blue represents a negative correlation. The bigger and darker diamond represents a higher correlation index. * represents *P*-value < 0.05, ** represents *P*-value < 0.01, *** represents *P*-value < 0.001.

*Granulicella*) had observed a positive correlation with two fungal genera (*Boothiomyces* and *Panaeolus*), bacterial genera *Bellilinea* and *Novosphingobium,* respectively, had significant negative correlation with fungal genera *Boothiomyces* and *Panaeolus*, *Coniochaeta* and *Pseudeurotium*, which had no significant correlation in *O. sativa*. On the contrary, in *O. sativa*, 10 bacterial genera (*Ellin 6067*, *Anaerolinea*, *Desulfobacca*, *Occallatibacter*, *MND1*, *Bryobacter*, *Candidatus Solibacter*, *Spirochaeta*, *Geothrix,* and *Haliangium*), respectively, had observed positive correlation with eight fungal genera (*Westerdykella*, *Clitopilus*, *Metapochonia*, *Pyrenochaetopsis*, *Sarocladium*, *Microdochium*, *Clonostachys,* and *Pyrenochaetopsis*), both *Ellin* and *Geothrix* were observed positive correlation to *Westerdykella*, and both *Spirochaeta* and *Haliangium* were observed positive correlation to *Pyrenochaetopsis*), six bacterial genera (*Nitrospira*, *Occallatibacter*, *Spirochaeta*, *Bellilinea*, *Geothrix,* and *Anaeromyxobacter*) had significant negative correlation with five fungal genera (*Nigrospora*, *Clitopilus*, *Westerdykella*, *Edenia,* and *Pyrenochaetopsis*, both *Nitrospira* and *Geothrix* were significant correlation to *Nigrospora*, both *Occalatibacter*

and *Spirochaeta* were significant correlation to *Clitopilus*, and *Anaeromyxobacter* was significant correlation to both *Edenia* and *Pyrenochaetopsis*), which had no significant correlation in *O. rufipogon* (Fig. S4A and B).

## Analysis of core microorganisms in genus between *O. rufipogon* and *O. sativa*

We defined a bacterial or fungal genus as the core microorganism based on two standards, which accounted for more than 1% relative abundance in the whole bacterial or fungal microorganisms and existed in more than 90% of *O. rufipogon* or *O. sativa* samples. As shown in Fig. S3; Fig. 5, core bacteria of *O. rufipogon* and *O. sativa* were, respectively, consisted of 30 and 40 genera within 26 common ones which contained two significant relative higher abundance of genera (*Bryobacter* and *Bradyrhizobium*) in *O. rufipogon* and three significant relative higher ones (*MND1*, *Nitrospira,* and *Ellin6067*) in *O. sativa*, while 4 and 14 specific genera, respectively, existed in the two *Oryza* species that contained two (*Denitratisoma* and *Opitutus*) and 13 (*Pseudomonas*, *Acinetobacter*, *Stenotrophomonas*, *Duganella*, *Sphingobacterium*, *Dyella*, *Bdellovibrio*, *Terrimonas*, *Aquicella*, *Massilia*, *Pelobacter*, *Sphingomonas,* and *Thiobacillus*) significant different genera (Fig. 5A). For fungus, core fungi of *O. rufipogon* and *O. sativa* were, respectively, consisted of 21 and 23 genera within 15 common ones which contained one significant relative higher abundance of genera (*Clitopilus*) in *O. rufipogon* and two significant relative higher ones (*Clonostachys* and *Conocybe*) in *O. sativa*, while six and eight specific genera, respectively, existed in the two *Oryza* species that contained two (*Hebeloma* and *Neurospora*) and seven (*Echria*, *Sarocladium*, *Parasarocladium*, *Tricoderma*, *Zopfiella*, *Metapochonia,* and *Limnoperdon*) significant different genera (Fig. 5B).

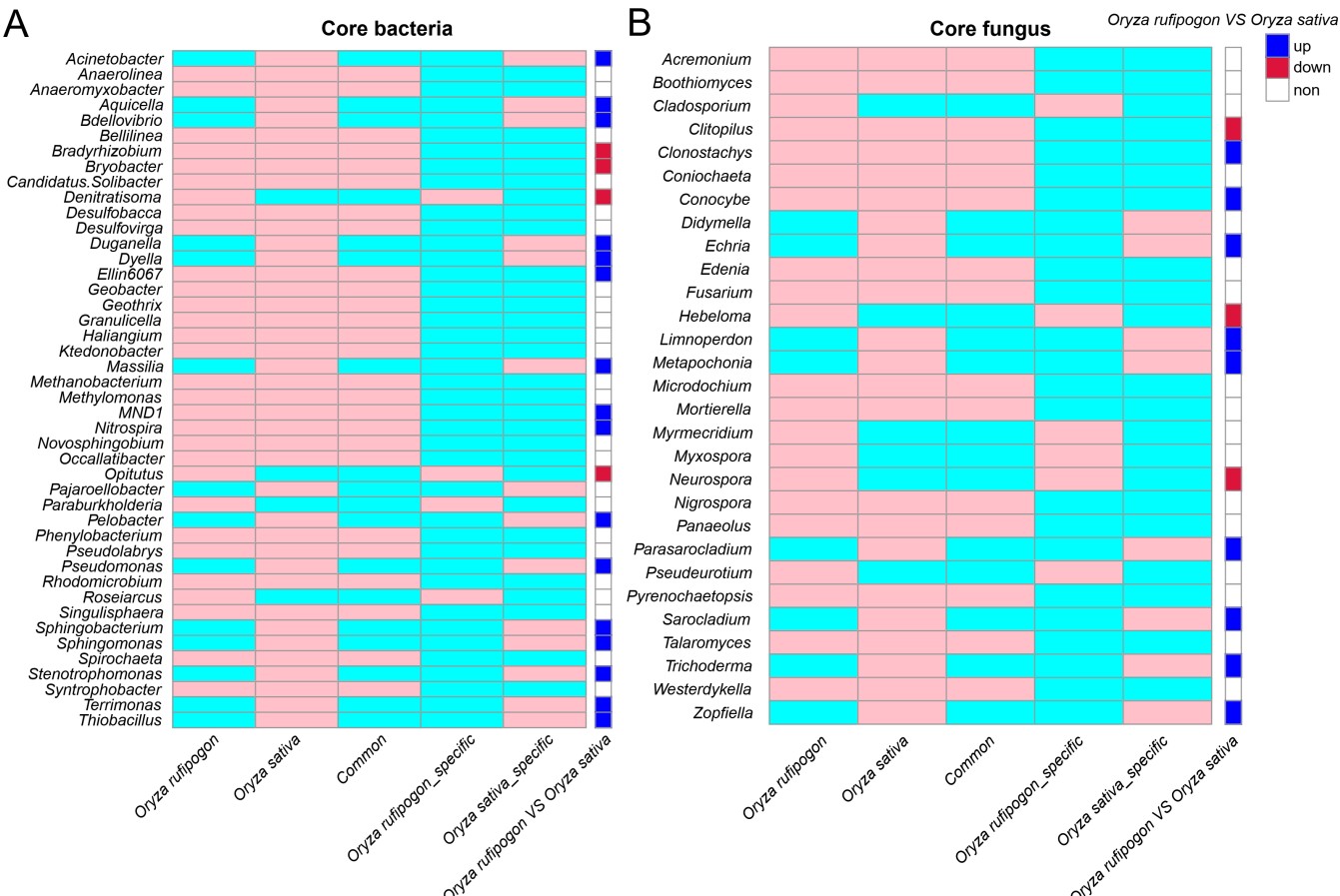

**FIG 5** Core bacterial and fungal community composition at the genera level of *O. rufipogon* and *O. sativa*. (A) Bacteria. (B) Fungi.

## Correlation analysis of significantly different agronomic traits and microorganisms between *O. sativa* and *O. rufipogon*

The common wild rice (*O. rufipogon*) is the ancestor of cultivated rice (*O. sativa*), with significant differences in many important agronomic traits such as leaf sheath color (most wild rice are purple while most cultivated rice are green), growth habit (most wild rice creep while most cultivated rice upright), awn (most wild rice have long awn while most cultivated rice have short awn or awnless), ligule color (most wild rice are purple while most cultivated rice are green), and height (most wild rice are tall while most cultivated rice are dwarf).

As shown in Fig. 6A, the relative abundance of *Bradyrhizobium*, *Opitutus*, *Bryobacter*, and *Denitratisoma* in *O. rufipogon* was significantly higher than in *O. sativa*, indicating that the four bacterial genera had an obvious positive correlation with a difference of phenotypes of *O. rufipogon* compared with *O. sativa*, including purple ligule and leaf sheath, long awn, procumbent growth, and tall plants, except for the non-significant correlation between *Denitratisoma* and purple leaf sheath. Oppositely, four bacterial genera had a significant negative correlation with five agronomic traits of *O. rufipogon* above, such as *Terrimonas*, *Thiobacillus*, *MND1,* and *Nitrospira*.

As shown in Fig. 6B, the relative abundance of *Clitopilus* and *Neurospora* in *O. rufipogon* were significantly higher than in *O. sativa*, indicating that the two fungal genera had an obvious positive correlation with a difference of phenotypes above of *O. rufipogon* compared with *O. sativa*. Oppositely, four fungal genera had a significant negative correlation with the five agronomic traits of *O. rufipogon* above, such as *Conocybe*, *Zopfiella*, *Trichoderma,* and *Limnoperdon*.

## DISCUSSION

Plant root microbiota is closely associated with plants, which have important roles in regulating many important life processes of plants (3, 40, 47), such as growth, development, immunity, and so on. Previous research in rice (27–29) indicated that not only

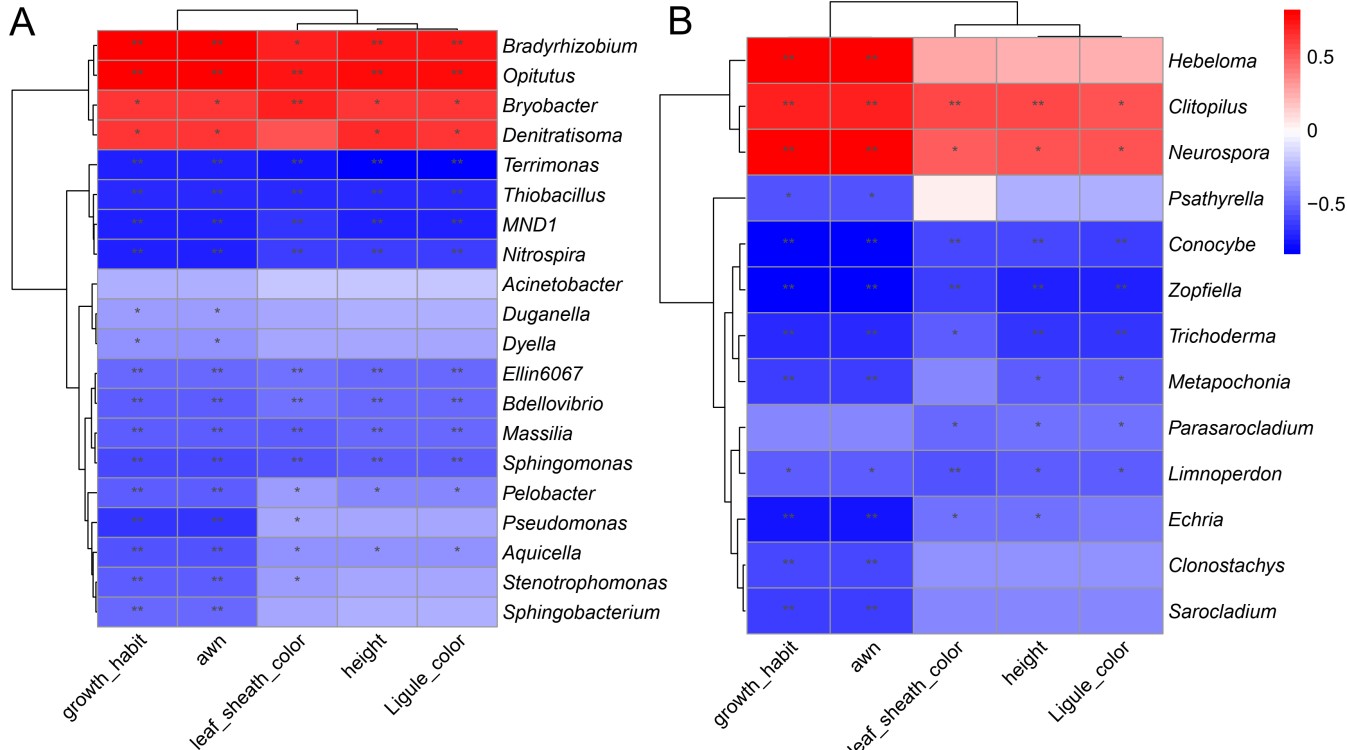

**FIG 6** Correlation analysis of significantly different agronomic traits and microorganisms between *O. sativa* and *O. rufipogon*. (A) Bacteria. (B) Fungi.

plant species and genotype but also soil microbiota, soil physicochemical properties, and plant secondary metabolites could be influential in the composition of rhizosphere microbe. As is well known, cultivated rice is domesticated from its ancestor, the common wild rice, for why *O. sativa* and *O. rufipogon* have many differences in various agricultural traits (growth habit, awn, leaf sheath color, and so on) and biotic (pathogen and pest) and abiotic resistances (cold, hot, salty, and so on). In this study, rice root microbial community diversity and structure between the two *Oryza* species were examined via high-throughput sequencing of ITS and 16S rRNA genes.

Based on the analysis of the abundance of microorganisms, we found that for bacteria, *Proteobacteria* and *Bacteroidetes* had relatively higher abundance in *O. sativa*, the abundance of *Thaumarchaeota* was relatively higher in *O. rufipogon*, while for fungi, *Chytridiomycota* and *Basidiomycota* had relatively higher abundance in *O. sativa*, the abundance of *Ascomycota* was relatively higher in *O. rufipogon*. Although the abundance of each fungal and bacterial phylum is different between *O. rufipogon* and *O. sativa*, the dominant phyla in the two *Oryza* species are always similar, such as *Proteobacteria*, *Thaumarchaeota*, *Bacteroidetes*, *Chloroflexi,* and *Acidobacteria* were the five dominant bacterial phyla, while *Ascomycota*, *Basidiomycota*, *Chytridiomycota*, *Mortierellomycota*, and *Rozellomycota* were the five dominant fungal phyla. This result was similar to other plants, such as barley (48), cotton (49), *Arabidopsis* (50), and rice (51, 52).

In our research, all three indexes (Chao1, Observed species, and PD whole tree) <0.05 showed a significantly higher bacterial diversity in *O. sativa* than in *O. rufipogon*, while only PD whole tree index <0.05 indicated an obviously lower fungal diversity in *O. sativa* compared to *O. rufipogon* based on alpha diversity analysis. However, relatively higher fungal alpha diversity occurred in the cultivated varieties than in the wild accessions both in soybean (53) and maize (54), even in rice (51), which demonstrated that both the fungal and the bacterial diversity in *O. sativa* were relatively higher than in *O. rufipogon* based parent-child relationship between the two *Oryza* species. The wild rice accessions and the cultivated varieties in our research didn't have a direct parent-child relationship may be the cause of relatively higher fungal abundance in *O. rufipogon* and lower in *O. sativa*. The PCoA analysis of fungi and bacteria based on Bray–Curtis showed distinct two groups of *O. rufipogon* and *O. sativa* in our study, which is similar to other reports (51, 55, 56).

Based on the correlation analysis of bacterial and fungal microorganisms within different abundance between *O. sativa* and *O. rufipogon*, we found that most microorganisms that had an obvious negative or positive correlation with other microorganisms existed either in *O. sativa* or in *O. rufipogon*, and only a few showed the same positive or negative correlation both in the two *Oryza* species. This result indicated that not only the composition of root microbiota but also the interaction among them was significantly different between *O. rufipogon* and *O. sativa*. A lot of research has demonstrated that rice root microbiota was significantly affected by a long time of domestication from *O. rufipogon* to *O. sativa* (31, 51, 55, 56). *Anaerolinea*, *Anaeromyxobacter,* and *Bradyrhizobium* were identified to be nitrogen-fixing bacteria (57), which were the common core bacteria in the two *Oryza* species and showed the relative abundance of *Bradyrhizobium* was significantly higher in *O. rufipogon* than in *O. sativa*, although the abundance of the other two bacterial genera had no obvious difference in the two *Oryza* species, indicating that the wild rice had a more reasonable strategy for utilizing nitrogen by root nitrogen relative microorganisms to response to the environment stress.

Based on the core microbiota analysis and the correlation analysis of significantly different agriculture traits and microorganisms in *O. rufipogon* and *O. sativa*, we found that although significant changes in root microbiota had occurred during a long time of domestication (58), most core microorganisms were retained with relative stable abundance while only a few were lost or newly recruited. Interestingly, most of these changed microorganisms had a significant high positive or negative correlation with at least one domestication trait, indicating that rice root microbiota selectively discarded or recruited special microorganisms to adapt to the changing environment, including

carbohydrate metabolism, cofactors and vitamin metabolism, transcription, replication, and repair (Fig. S5).

As is well known, during the process of cultivating rice varieties domesticated from wild rice accessions, many beneficial characteristics were lost, especially the resistance to adversity including pathogens, pests, cold, hot, drought, and so on, meanwhile, the root microbiota also had been obviously affected. Many root microorganisms have demonstrated that they can effectually promote host plants absorbing nutrition and responding to unfriendly environments for survival and development. It is of great significance to improve the rice quality by digging and utilizing root microbial resources from the wild rice accessions. The results in our study systematically reflected the significance of root microbiota between *O. rufipogon* and *O. sativa*, providing a theoretical basis for improving the rice-cultivated varieties through an angle of interaction between root microbiota and host plants.

## ACKNOWLEDGMENTS

This work was financially supported by grants from the Guangzhou Science Technology and Innovation Commission (202102021007) and the Guangdong Key Laboratory of New Technology in Rice Breeding (2023B1212060042).

L.C. and D.Y.Z. designed the experiments and wrote the paper together with J.L.Q. and K.D.; C.W.F. and F.Z.L. planted the rice plants and investigated the phenotypes; S.B.R. and Z.J. performed preparation of root samples; H.L. and Y.S.J. performed the extraction of DNA and amplification of ITS and 16S rRNA gene; L.S.W., Y.H., C.P.L., M.X.X., and L.Q., respectively, performed alpha diversity analysis, principal coordinates analysis, core microorganisms analysis, correlation analysis among microorganisms, and correlation analysis between phenotypes and microorganisms.

## AUTHOR AFFILIATIONS

[1]Rice Research Institute, Guangdong Academy of Agricultural Sciences, Guangdong Key Laboratory of New Technology in Rice Breeding, Guangdong Rice Engineering Laboratory, Guangzhou, China

[2]Healthtimegene Institute, Shenzhen, China

[3]State Key Laboratory for Conservation and Utilization of Subtropical Agro-Bioresources, Guangdong Province Key Laboratory of Microbial Signals and Disease Control, Integrative Microbiology Research Centre, South China Agricultural University, Guangzhou, China

## AUTHOR ORCIDs

Liqun Jiang http://orcid.org/0000-0001-6821-9779
Yizhen Deng http://orcid.org/0000-0002-3572-1559
Chen Li http://orcid.org/0000-0001-6702-6860

## FUNDING

| Funder | Grant(s) | Author(s) |
|---|---|---|
| Guangzhou Science Technology and Innovation Commission | 202102021007 | Liqun Jiang |
| Guangzhou Science Technology and Innovation Commission | 2023B1212060042 | Liqun Jiang |

## AUTHOR CONTRIBUTIONS

Liqun Jiang, Funding acquisition, Writing – original draft | Da Ke, Writing – original draft | Bingrui Sun, Resources | Jing Zhang, Resources | Shuwei Lyu, Methodology | Hang Yu, Data curation | Pingli Chen, Methodology | Xingxue Mao, Methodology | Qing

Liu, Methodology | Wenfeng Chen, Resources | Zhilan Fan, Resources | Li Huang, Data curation | Sanjun Yin, Data curation | Yizhen Deng, Supervision, Writing – review and editing | Chen Li, Supervision, Writing – review and editing

## ADDITIONAL FILES

The following material is available online.

### Supplemental Material

**Figure S1 (Spectrum03330-23-s0001.pdf).** Relative abundance of bacterial and fungal composition at genus level of *Oryza rufipogon* and *Oryza sativa*.
**Figure S2 (Spectrum03330-23-s0002.pdf).** Venn diagram showing differences in bacterial and fungal community composition at OTU level of *Orza rufipogon* and *Oryza sativa*.
**Figure S3 (Spectrum03330-23-s0003.pdf).** Venn diagram showing differences in core bacterial and fungal community composition at genus level of *Oryza rufipogon* and *Oryza sativa*.
**Figure S4 (Spectrum03330-23-s0004.pdf).** Spearman correlation of bacteria-fungi at genus level in *Oryza rufipogon* and *Oryza sativa*.
**Legends (Spectrum03330-23-s0005.docx).** Legends to Fig. S1 to S4.
**Table S1 (Spectrum03330-23-s0006.pdf).** Sample information.
**Table S2 (Spectrum03330-23-s0007.pdf).** Summary of the investigated samples in the current study.

### Open Peer Review

**PEER REVIEW HISTORY (review-history.pdf).** An accounting of the reviewer comments and feedback.

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
