## [Reviewer comments · Microbiology Spectrum]

Microbiology Spectrum

Root microbiota analysis of *Oryza rufipogon* and *Oryza sativa* reveals an orientation selection during the domestication process

Liqun Jiang, Da Ke, Bingrui Sun, Jing Zhang, Shuwei Lyu, Hang Yu, Pingli Chen, Xingxue Mao, Qing Liu, Wenfeng Chen, Zhilan Fan, Li Huang, Sanjun Yin, Yi Zhen Deng, and Chen Li

Corresponding Author(s): Chen Li, Rice Research Institute, Guangdong Academy of Agricultural Sciences

Review Timeline:

Submission Date:

September 10, 2023

Accepted:

February 15, 2024

Editor: Zhongxiong Lai

Reviewer(s): Disclosure of reviewer identity is with reference to reviewer comments included in decision letter(s). The following individuals involved in review of your submission have agreed to reveal their identity: ERKAY ÖZGÖR (Reviewer #3)

Transaction Report:

DOI: <https://doi.org/10.1128/spectrum.03330-23>

Re: Spectrum03330-23 (Root microbiota analysis of *Oryza rufipogon* and *Oryza sativa* reveals an orientation selection during the domestication process)

Dear Prof. Chen Li:

my decision is accept.

Your manuscript has been accepted, and I am forwarding it to the ASM production staff for publication. Your paper will first be checked to make sure all elements meet the technical requirements. ASM staff will contact you if anything needs to be revised before copyediting and production can begin. Otherwise, you will be notified when your proofs are ready to be viewed.

Sincerely,
Zhongxiong Lai
Editor
Microbiology Spectrum

Reviewer #2 (Comments for the Author):

In this manuscript, the authors comparatively analyzed the difference of root microbiome of wild rice *Oryza rufipogon* and cultivated rice *Oryza sativa* through high-throughput amplicon sequencing and bioinformatic analysis. And the results demonstrated a strong correlation between the core microorganisms of the two *Oryza* species and their agricultural traits. The findings will contribute to rice breeding from the perspective of interactions between root microorganisms and plants.

Major concerns:

1. As author mentioned when compared to the cultivated rice, the wild rice exhibits different agricultural traits, including an improved ability to survive in response to biotic and abiotic stress. The correlation between the root microbiomes of the two *Oryza* species and their agricultural traits was also investigated. However, disease resistance as an important agricultural trait was not analyzed and no relationship with root microbiomes was showed. Therefore, I recommend that relevant content be added to the text.
2. I guess the high-throughput sequencing of this experiment is done by the company, but it is not reflected in the materials and

methods, which should be completed.

3. Furthermore, redundancy analysis (RDA) of correlation should not be showed in the statistical test. And RDA should be described in detail.

4. In statistical test, the procedure for specific difference test and the parameter should be supplemented and clearly defined.

5. Line 71 and 85, when *Oryza rufipogon* was presented for the second time, it could be abbreviated as *O. rufipogon*. Please double-check the full text.

Reviewer #3 (Comments for the Author):

The manuscript titled "Root microbiota analysis of *Oryza rufipogon* and *Oryza sativa* reveals an orientation selection during the domestication process" is well-written and shows better results in its figures. There is no complexity in the whole explanation.